# Infection-Related Ventilator-Associated Complications in Critically Ill Patients with Trauma: A Retrospective Analysis

**DOI:** 10.3390/antibiotics12010176

**Published:** 2023-01-15

**Authors:** Emanuele Russo, Marta Velia Antonini, Andrea Sica, Cristian Dell’Amore, Costanza Martino, Emiliano Gamberini, Luca Bissoni, Alessandro Circelli, Giuliano Bolondi, Domenico Pietro Santonastaso, Francesco Cristini, Luigi Raumer, Fausto Catena, Vanni Agnoletti

**Affiliations:** 1Anesthesia and Intensive Care Unit, Bufalini Hospital, AUSL Romagna, 47521 Cesena, Italy; 2Department of Biomedical, Metabolic and Neural Sciences, University of Modena & Reggio Emilia, 41121 Modena, Italy; 3Anesthesia and Intensive Care Unit, Umberto I Hospital, AUSL Romagna, 48022 Lugo, Italy; 4Anesthesia and Intensive Care Unit, Infermi Hospital, AUSL della Romagna, 47923 Rimini, Italy; 5Infectious Diseases Unit, Forlì-Cesena Hospitals, AUSL Romagna, 47121 Forlì-Cesena, Italy; 6Department of Emergency Surgery and Trauma, Bufalini Hospital, AUSL Romagna, 47521 Cesena, Italy

**Keywords:** ventilator associated pneumonia, trauma, traumatic brain injury, mechanical ventilators, antimicrobial stewardship, anti-infective agents, intensive care unit

## Abstract

Background: Trauma is a leading cause of death and disability. Patients with trauma undergoing invasive mechanical ventilation (IMV) are at risk for ventilator-associated events (VAEs) potentially associated with a longer duration of IMV and increased stay in the intensive care unit (ICU). Methods: We conducted a retrospective cohort study aimed to evaluate the incidence of infection-related ventilator-associated complications (IVACs), possible ventilator-associated pneumonia (PVAP), and their characteristics among patients experiencing severe trauma that required ICU admission and IMV for at least four days. We also determined pathogens implicated in PVAP episodes and characterized the use of antimicrobial therapy. Results: In total, 88 adult patients were included in the main analysis. In this study, we observed that 29.5% of patients developed a respiratory infection during ICU stay. Among them, five patients (19.2%) suffered from respiratory infections due to multi-drug resistant bacteria. Patients who developed IVAC/PVAP presented lower total GCS (median value, 7; (IQR, 9) vs. 12.5, (IQR, 8); *p* = 0.068) than those who did not develop IVAC/PVAP. Conclusions: We observed that less than one-third of trauma patients fulfilling criteria for ventilator associated events developed a respiratory infection during the ICU stay.

## 1. Introduction

Despite high variabilities associated with country, age, and sex, trauma represents a leading external cause of death and disability [1]. Timely recognition of trauma severity, appropriate treatment, and post-resuscitation care could deeply impact the prognosis and patient-sensitive outcomes in this population [1].

Management of major trauma, inconsistently defined by current literature [2,3,4,5,6,7], frequently requires intensive care unit (ICU) admission. The implementation of invasive strategies to provide life support, such as endotracheal intubation and invasive mechanical ventilation (IMV), could increase the risk of healthcare-related infectious complications among this population [8,9].

Globally, patients with severe trauma present an increased risk of developing infections due to several physiopathological reactions to the presenting insult, such as tissue hypoperfusion, reduced immunological competence, and excessive cytokine release [9,10]. Traumatic injuries involving the head, thorax, and abdomen could impact the respiratory mechanism, thereby increasing the infection risk [11]. Lower respiratory tract infections, including hospital-acquired pneumonia, could be particularly frequent in patients presenting with traumatic brain injury (TBI), with more severe cerebral trauma, associated thoracic injury, and aspiration among known risk factors [12]. Respiratory infections and pneumonia can increase the need for tracheostomy and impair outcomes in patients with TBIs [12,13].

The diagnosis of pulmonary infection in mechanically ventilated patients features critical challenges [14]. The Centers for Disease Control and Prevention (CDC) developed a new paradigm for the classification and monitoring ventilator-associated events (VAEs) including infectious complications, categorized as infection-related ventilator-associated complication (IVAC), and possible ventilator-associated pneumonia (PVAP) [15,16,17].

VAE episodes have been associated with prolonged MV [18] and increased length of stay [18,19]; however, an independent relationship between ICU outcomes and mortality has not been established [18,20] despite some authors reporting an association with specific populations of patients with trauma [21]. In studies conducted prior to introducing the surveillance paradigm, attributable mortality close to zero has been reported for ventilator-associated pneumonia (VAP) in patients with trauma [22], even when VAP was reported as the most common nosocomial infection in trauma associated with prolonged MV and hospital stay, related systemic infectious complications, elevated costs [23], and reduced likelihood of discharge [22].

Additional challenges include the coexistence of multiple triggers for infectious respiratory complications [24]. The frequent presence of findings, such as fever, potentially mimicking patient symptoms [25,26] can further impair diagnosis and decision-making concerning the appropriate time to initiate antimicrobial therapy.

In this paper, we report data from a retrospective cohort study on patients experiencing severe trauma requiring ICU admission and MV; aim to evaluate the incidence and the characteristics of IVACs and PVAPs in this population; and investigate the implicated pathogens and the use of antimicrobial therapy.

## 2. Materials and Methods

We conducted a retrospective analysis of anonymous electronic health records, including consecutive patients admitted to the 18-bed ICU of Bufalini Hospital, AUSL della Romagna (Level I Trauma Center, TC within the “Romagna” subregional SIAT) [27], between 1 January 2019 and 31 December 2019. The Ethical Committee of the AUSL della Romagna (Comitato Etico della Romagna, C.E. Rom.) reviewed and approved data entry in selected ICUs and trauma registries, as well as their use for retrospective research purposes. The study adhered to the guidelines of the Declaration of Helsinki (1975) [28].

All patients admitted to our ICU during the study period and meeting the following criteria were deemed eligible for inclusion in the main analysis:-≥16 years of age;-Primary diagnosis of trauma;-MV requirement ≥ four consecutive days.

We collected and analyzed demographic data and descriptors characterizing the severity of trauma and associated neurological impairment and injuries, specifically the GCS [29,30], TBI severity [31], ISS [6,7], and the abbreviated injury scale (AIS) [1,4,5].

The occurrence of VAE tiers, IVACs, and PVAPs, was assessed according to the 2022 updated CDC definition [16]. Ventilator-Associated Condition(VACs) are defined as the deterioration of respiratory status after a period of stability or improvement. The subsequent tiers of VAEs recognize infectious conditions. IVACs require, on or after day 3 of IMV and within 2 days before or after worsening oxygenation, the concurrent presence of abnormal temperature or white cell count in patients receiving one or more new antibiotics for at least 4 days. PVAP refers to the presence of IVAC with evidence of purulent secretions and pathogenic cultures alone or along with positive pleural fluid cultures, positive lung histopathology, diagnostic tests for Legionella spp., or other selected viruses [14,16]. Data concerning patients suffering from IVAC and PVAP were aggregated in the analysis process.

As per standard practice in the ICU conducting the study, a consistent multimodal monitoring strategy was fulfilled for all patients admitted for early detection of potential infectious pulmonary complications. This strategy included continuous measurement and recording of central body temperature and daily white blood cell counts. A lung ultrasound was performed at least daily, and chest radiography was performed if any complication was suspected. 

The appearance (thickness and color) of sputum or tracheobronchial secretions was examined and recorded if spontaneous cough and/or tracheobronchial aspiration was noted, given that their presence was clinically suspected or detected. If an ongoing infection was suspected, quantitative cultures of respiratory secretions (on tracheobronchial aspirate or bronchoalveolar lavage through fiberoptic bronchoscopy) were performed and repeated serially until the initiation of antibiotic therapy. If a patient presents with risk factors for multidrug-resistant (MDR) bacteria, quantitative cultures were performed at ICU admission. Antibiotic therapy was prescribed as deemed necessary based on clinical judgment. If culture test results were available when antibiotics were initiated, targeted therapy was administered; empirical therapy was initiated if no microbiological data were available. No antibiotic prophylaxis was administered to prevent aspiration pneumonia or IVAC/PVAP in patients with TBI, given that no currently available evidence justifies this practice [32,33]. Antibiotic prophylaxis was administered in the case of exposed associated fractures or if any surgical intervention was required.

The primary outcome was the incidence of IVACs and PVAPs. The secondary outcomes were the time from ICU admission to IVAC/PVAP, the incidence of resistant pathogens (defined consistently with the currently accepted definitions proposed for characterizing non-susceptibility in bacterial isolates) [34]; total duration of antibiotic treatment in this specific population; the proportion of patients who received empirical antibiotics; the proportion of patients who received culture-directed antibiotic treatment; IMV duration; and length of ICU stay. We compared demographic data, severity scores, and neurological impairment scores at ICU admission between patients with and without IVAC/PVAP.

Descriptive statistics were performed to report demographic and clinical data. Quantitative variables are reported as median values (interquartile range (IQR)) and mean (standard deviation (SD)), whereas qualitative variables are reported as numbers (absolute and %). Statistical analyses were performed using the Mann–Whitney U and χ^2^ tests. The relative risk of TBI patients developing IVAC/PVAP was calculated according to Deeks et al. [35].

The RECORD statement for reporting observational studies conducted using routinely collected health data was followed [36].

## 3. Results

In total, 246 adult patients (≥16 years old) required ICU admission owing to major traumatic injuries during the study period. Overall, 144 patients (58.5%) with an ICU length of stay (LoS) below 96 h (four days) were excluded as minimal LoS for the potential development of VAEs was not met. Among the 102 adult patients (41.5%) exhibiting an ICU LoS ≥ 96 h, 13 were excluded from the main analysis. Specifically, 1 patient (0.4%) did not undergo IMV, while 12 patients (4.9%) were managed with IVM for <96 h, thereby not fulfilling the established criteria for the potential development of VAEs.

Finally, 88 adult patients were included in the main analysis (Figure 1).

Among the 88 selected cases, 3.4% of patients (n = 3) developed specific signs of infection without consistent identification of the site and causative organisms. In addition, 4.5% of patients (n = 4) developed non-respiratory-related infections: 3.4% (n = 3) developed skin and soft-tissue infections; 2.3% (n = 2) developed peritonitis (one patient developed both skin and soft-tissue infection and peritonitis). Furthermore, 29.5% (n = 26) of patients who fulfilled the criteria for the potential development of ventilator-associated events developed respiratory infections during their ICU stay (10.6% of the total adult patients admitted for trauma during the study period). Out of these, 24 (92.3%) patients met the criteria for PVAP.

Baseline characteristics, trauma-associated injury severity, and ICU-associated treatments of the overall population stratified according to pulmonary infection status are summarized in Table 1. Data related to patients suffering from IVAC and PVAP were aggregated.

The median age distribution was similar between patients who developed IVAC/PVAP and those who did not (55.5 years (IQR 29.5) vs. 59.5 (IQR 24.5), respectively). Overall, 19.2% (n = 5) of patients who developed IVAC/PVAP were female, compared with 27.4% (n = 17) who did not develop IVAC/PVAP.

Patients who developed IVAC/PVAP presented lower total GCS (mean value 8.6 ± 4.7 vs. 10.6 ± 4.2; median value, 7; IQR, 9 vs. 12.5, IQR 8; *p* = 0.068) than those who did not develop IVAC/PVAP. 

Moreover, these patients exhibited an elevated ISS (mean value 39.6 ± 11.0 vs. 35.3 ± 15.3; median value 41, (IQR 16) vs. 34 (IQR 16); *p* = 0.040).

Compared with patients with mild/moderate TBI (GCS > 9), patients with severe TBI (GCS 3–8) demonstrated an increased risk of developing IVAC/PVAP (relative risk 1.68; 95% CI 0.89–3.21).

In addition, AIS for the abdomen (AIS a) was elevated (mean value 2.0 ± 2.0 vs. 1.1 ± 1.9; median value 2 (IQR) 4 vs. 0 (IQR 3), *p* = 0.021), whereas AIS for other regions did not significantly differ between patients in the two groups.

Patients developed IVAC/PVAP following a minimum three-day ICU stay to a maximum of 13 days. The mean and median times required for IVAC/PVAP development were 6.15 (SD, ± 2.74) and 6 days (IQR 3), respectively. Based on definitions included in the updated CDC surveillance paradigm, 2 patients developed IVACS, and 24 patients developed PVAP. Among them, two patients developed pleural empyema, A summary of the relative percentages is shown in Figure 2. 

Compared with patients who did not develop infectious VAEs, those who developed IVAC/PVAP presented an elevated ICU LoS (mean LoS 16.7 ± 7.1 days vs. 9.0 ± 8.7 days; median LoS 16.5, IQR 8 days vs. 7, IQR 5 days; *p* < 0.001) and increased MV duration (mean 15.7 ± 6.4 days vs. 8.0 ± 8.3 days; median 15, IQR 9 days vs. 8, IQR 8 days; *p* < 0.001). Moreover, 53.8% (n = 14) of patients who developed IVAC/PVAP required tracheostomy when compared with 17.7% (n = 11) of patients who did not develop IVAC/PVAP (*p* < 0.001).

Five patients, i.e., 19.2% of those presenting with IVAC/PVAP and 5.6% of the whole study group, exhibited two IVAC/PVAP episodes during the ICU stay.

Overall, 19 different pathogens were isolated from the respiratory tract of patients with IVAC/PVAP. 

Among pathogens implicated in the IVAC/PVAP population, *Staphylococcus aureus*, *Escherichia coli*, and *Hemophilus influenzae* were the most frequently detected, with 23 (25.9%), 13 (16%), and 10 (12.3%) positive samples, respectively, representing a total of 81 isolates. The frequency of isolated bacteria is presented in Figure 3.

Five patients (19.2%) were diagnosed with respiratory infections due to resistant bacteria. Specifically, considering bacterial isolates, 73 (90.1%) non-resistant organisms and seven (8.6%) multi-drug resistant (MDR) organisms were documented; one extensively drug-resistant (XDR) organism was detected (1.2%), with no pandrug-resistant (PDR) organism observed. Additionally, no carbapenem-resistant Enterobacterales were isolated from the study group.

Nine patients (34.6%) did not receive empirical antibiotic therapy, and three (11.5%), seven (26.9%), six (23.1%), and one (3.8%) patients received 1, 2, 3, and >3 (7 days) days of empirical antibiotic therapy, respectively.

Considering culture-directed therapy, two patients (7.7%) did not receive this therapy, two (7.7%) received therapy for two days, five (19.2%) received three days of therapy, seven (26.9%) received four days of therapy, four4 (15.4%) received five and six days of therapy, and two patients received seven and eight days of antibiotic therapy (3.8% of patients at each time point).

No signs of systemic involvement were observed in 23 (88.5%) patients. Three patients (11.5%) developed septic shock, as defined by the Third International Consensus Definitions for Sepsis and Septic Shock (Sepsis-3) [37,38,39]. All three patients presented with at least one focus (in two cases) or multiple (one patient) additional foci of infection.

## 4. Discussion

In this study, we performed a retrospective analysis assessing the development of IVACs and PVAPs in a population of patients with trauma and reported the incidence of these complications by rigorously applying the updated CDC criteria [16]. Limitations and issues associated with diagnosing infections as potentially related to MV, regardless of the adopted criteria, are widely recognized [14,15,16,17,18,19,20,21,22,23,24,25,26,27,28,29,30,31,32,33,34,35,36,37,38,39,40].

The trauma-induced hyperinflammation, mimicking signs of infection [26], and the possible presence of trauma-associated multiple foci of infections may further complicate the scenario. Moreover, trauma-related chest injuries can impair the assessment of radiological data. Detection of pleural empyema, for example, could be related to the primary insult rather than develop as a consequence of MV.

It should be kept in mind that IVAC criteria inherently exclude early ventilator-associated pneumonia

Because the latter consideration, the distinction between IVAC and PVAP in the study population, is complex, we decided to report both without distinguishing between the two throughout our study.

Risk factors for MDR colonization/infection are well described in previous studies [41]. In this study, we highlight the acceptable rate of MDR, the low rate of XDR (detected only once during the study period), and the absence of both PDR and carbapenemase-producing pathogens. This finding could be considered an inherent characteristic of the patient population, characterized by a short pre-MV hospital stay, and has a low-risk profile for M/X/PDR pathogen colonization and infection. In addition, specific ICU characteristics could have impacted our findings. Specifically, most patients with trauma were directly admitted to the emergency department without previous treatment in any ward. Furthermore, we accurately defined the screening parameters and isolated patients if resistant bacteria were detected or if any contact with an infected patient was suspected. The ICU where the study was carried out is involved in an antibiotic stewardship program and also reports an acceptable overall rate of MDR colonization [33].

Strictly monitoring local microbial ecology and M/X/PDR rates is critical for guiding empirical antibiotic therapies [42]. Notably, the studies were conducted before the coronavirus disease (COVID-19) pandemic. Studies during the “COVID-19 era” have reported a significant evolution in both ICU microbial profiling and the incidence of infections [43,44,45].

The low incidence of systemic involvement in trauma patients with IVAC/PVAP represents an interesting finding and may positively affect outcomes. This observation is consistent with previously published results [22].

In our population, patients who developed IVAC/PVAP presented with low GCS at ICU admission (although the difference did not reach statistical significance), suggesting that microaspiration and/or the need for deep sedation to control intracranial pressure might play a role in the pathophysiology of pulmonary infectious complications. Patients with IVAC/PVAP also had significantly increased ISS and abdominal AIS with no differences detected in thoracic AIS. We speculate that higher ISS can be associated with greater trauma severity, along with the need for a prolonged hospital stay and enhanced cytokine release [9]. The lack of differences in thoracic AIS suggests that infection superimposed on lung regions involving traumatic contusion might not represent a pivotal factor.

In addition, we found a significant difference between MV duration and ICU LoS between patients who developed IVAC/PVAP and those who did not develop this complication. Both might represent risk factors for IVAC/PVAP or may be a consequence of IVAC/PVAP. Nevertheless, the limited sample size of this study hindered the performance of multivariate analysis to resolve this uncertainty.

We limited our study to patients who were admitted and treated in 2019. Since the conclusion of this limited observation period, new evidence has emerged regarding the approach to IVAC/PVAP, leading to advances in their management. The introduction of fast molecular microbiology tests has overturned the concepts of empirical therapy/targeted therapy. Moreover, an ongoing trend to avoid excessive and possibly unnecessary durations of antibiotic therapies can be observed [46,47,48].

## 5. Limitations

This study has several factors that limit the generalizability of the obtained results.

The first and main limitation is the retrospective and single-center design with a small sample size. A larger sample size would allow for multivariate analysis in order to attempt to correct for clinical covariates and ventilation exposure time. In addition, a larger sample size would also aid in evaluating the risk factors for IVAC/PVAP and the association between these complications and the clinical course of the patients.

Data were collected from a heterogeneous trauma population; the presence or absence of trauma-related infection foci (e.g., gastrointestinal involvement and/or open fractures) might have affected the need for antibiotics and the eventual development of resistance or systemic compromise (sepsis and septic shock).

Owing to organizational factors, some patients were transferred from the Level I TC, acting as a hub, to the ICUs of other facilities within the Romagna SIAT (also including three Level II TCs, acting as spokes), given that care provided in a Level 1 TC was no longer required [49,50]. This protocol is in full agreement with a regional deliberation [27] which has aimed to optimize and rationalize trauma management in patients since 2002.

At the time of decentralization, some patients were still on MV, which may have resulted in an underestimation of the long-term incidence of VAEs, as the criteria for development could have been fulfilled after transfer. Accordingly, some patients were discharged from the ICU with ongoing antibiotic treatment (specifically, 4 patients over the 26 developing VAEs, 15,4%). Data regarding the characterization of antimicrobial therapy (appropriateness, time on empiric and/or culture-directed antibiotic therapies, and total duration) may not accurately reflect the whole study group.

## 6. Conclusions

We conducted a retrospective analysis assessing patients experiencing severe trauma that required ICU admission and MV. We reported the incidence and the characteristics of IVACs and PVAPs in this population, investigating the implicated pathogens and the use of antimicrobial therapy.

Despite some limitations, we were able to describe the rate of infectious VAEs in a selected population, a result that may become a benchmark. 

The reported differences between patients who developed and those who did not develop infectious VAEs, particularly with referral to GCS and ISS, suggest some thought-provoking considerations over the pathophysiological determinants of IVAC/PVAP. Moreover, the low systemic involvement associated with IVAC/PVAP in patients who did not present with additional foci of infection could be considered relevant.

The observed interesting tendencies would require a larger population for confirmation and raise questions requiring further studies, including a larger sample size and multicentric design.

## Figures and Tables

**Figure 1 antibiotics-12-00176-f001:**
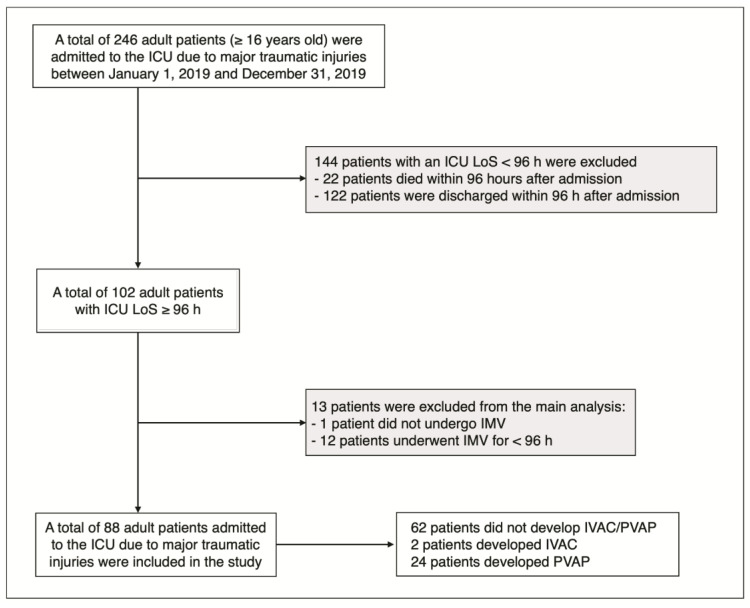
Flow chart of study population selection; gray boxes, subjects excluded from the main analysis (see methods). ICU: intensive care unit; LoS: length of stay; h: hours; IMV: invasive mechanical ventilation.

**Figure 2 antibiotics-12-00176-f002:**
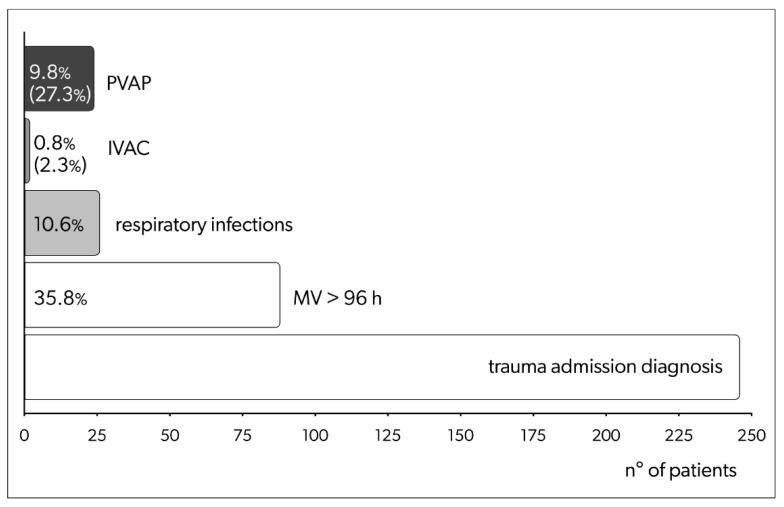
Patients admitted with trauma diagnoses during the study period; among these were patients on mechanical ventilation for more than 96 h (potentially fulfilling the criteria for IVAC of PVAP development), patients developing respiratory infection fulfilling criteria for VACs, and patients developing IVACs/PVAPs. For the last group, percentages are over total admissions (upper line) and over the number of individuals on MV for more than 96 h (lower line, between parenthesis). MV: mechanical ventilation; h: hours; IVAC: infection-related ventilator-associated complications; PVAP: possible ventilator-associated pneumonia; VAC: ventilator-associated conditions.

**Figure 3 antibiotics-12-00176-f003:**
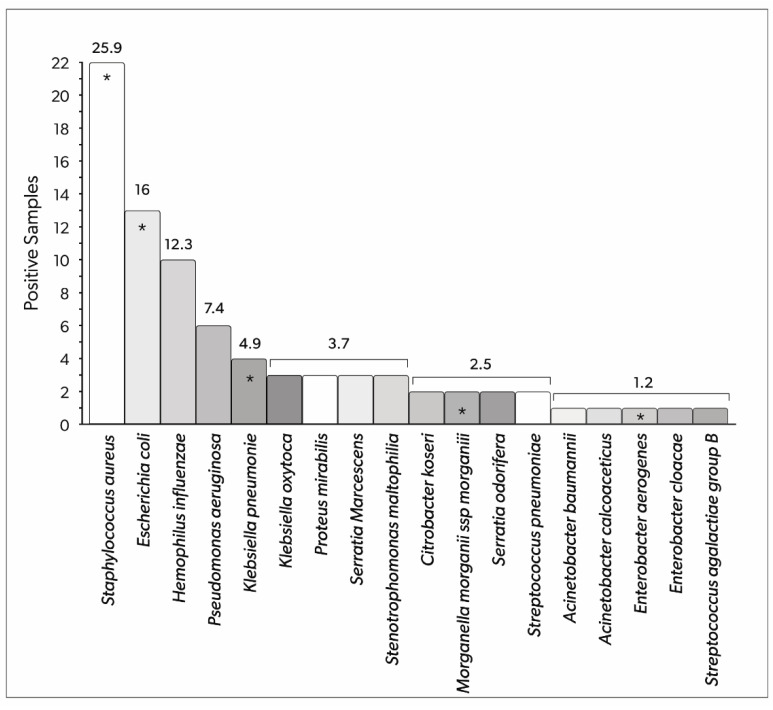
Bacteria detected in the respiratory tract samples of patients with IVAC/PVAP (total number of positive samples). The numbers above the columns represent the relative percentage of the total number of positive samples. Asterisks indicate pathogens that presented multi-drug resistance in at least one sample. IVAC: infection-related ventilator-associated complications; PVAP: possible ventilator-associated pneumonia.

**Table 1 antibiotics-12-00176-t001:** Baseline patient characteristics, trauma-associated injury severity, and ICU LoS/associated treatments of the overall population and stratified according to IVAC/PVAP status. IVAC: infection-related ventilator-associated complications; PVAP: possible ventilator-associated pneumonia; SD: standard deviation; IQR: interquartile range; BMI: body mass index; GCSt: Glasgow coma scale (total score); TBI: traumatic brain injury; ISS: injury severity score; AIS: abbreviated injury scale; c: chest; h: head; a: abdomen; s: skeletal; e: external; f: face; ICU: intensive care unit; LoS: length of stay.

		Total n = 88	NO IVAC/PVAP n = 62 (70.5%)	IVAC/PVAP n = 26 (29.5%)	*p*-Value *
Baseline characteristics					
Age (years)	mean (SD)	56.7 (±18.0)	58.68 (±17.2)	52.1 (±19.5)	0.137
Sex, male	n (%)	66 (75)	45 (72.6)	21 (80.8)	0.418
Trauma characteristics					
GCSt	median (IQR)	11 (8)	12.5 (8)	7 (9)	0.068
TBI	n (%)	88 (100)	62 (100)	26 (100)	0.218
mild TBI	n (%)	13 (14.8)	9 (14.5)	4 (15.4)	
moderate TBI	n (%)	39 (44.3)	31 (50)	8 (30.8)	
severe TBI	n (%)	36 (40.9)	22 (35.5)	14 (53.8)	
Pupils, abnormal	n (%)	15 (17.0)	9 (14.5)	6 (23.1)	0.330
Pupils, normal	n (%)	73 (83.0)	53 (85.5)	20 (76.9)	
ISS	median (IQR)	34 (17.5)	34 (16)	41 (16)	0.040
AIS t	median (IQR)	3 (4)	3 (4)	3 (4)	0.493
AIS h	median (IQR)	4 (3.5)	3,5 (4)	4 (2)	0.774
AIS a	median (IQR)	0 (3)	0 (3)	2 (4)	0.021
AIS s	median (IQR)	0 (3)	0 (3)	1 (3)	0.317
AIS e	median (IQR)	0 (1)	0 (1)	0 (1)	0.912
AIS f	median (IQR)	0 (3)	0 (3)	0 (3)	0.504
ICU characteristics/associated treatments					
ICU LoS (days)	median (IQR)	8.5 (10)	7 (5)	16.5 (8)	<0.001
MV duration (days)	median (IQR)	8 (8)	6 (5)	15 (9)	<0.001
Tracheostomy	YES n (%)	25 (28.4)	11 (17.7)	14 (53.8)	<0.001
	NO n (%)	63 (71.6)	51 (82.3)	12 (46.2)	
ABT prophylaxis *	YES n (%)	16 (18.2)	13 (21.0)	3 (11.5)	0.295
	NO n (%)	72 (81.8)	49 (79.0)	23 (88.5)	
Hemofiltration	n (%)	13 (14.8)	8 (12.9)	5 (19.2)	0.445

* Antibiotic prophylaxis was administered if deemed necessary due to open fractures and/or the need for surgery; no prophylaxis was administered to prevent IVAC/PVAP.

## Data Availability

Not applicable.

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
