# Peer review of "Infection-Related Ventilator-Associated Complications in Critically Ill Patients with Trauma: A Retrospective Analysis"

_antibiotics, 2023, doi:10.3390/antibiotics12010176_

Round 1

Reviewer 1 Report

The author performed a retrospective analysis of infection-related ventilator-associated complications in critically ill patients with trauma

comments

#1 Abstract

why is the independent relatioship between ICU outcomes and mortality highlighted in the abstract since this is not the main aim of this report

#2 please state in the results (abstract) all the statistical  significant results you found

#3 The conclusions (abstract) are too vague. Please state explicitly

#4 line 65-66 should combine with eprevious paragraph

#5 line97-99 should combine with the previous paragraph

#103-110 please state in this paragraph that you combined PVAP with IVAC

#131-132 please combine with previous paragraph

# 163-170 Do the total number of patients described make the total of n=41?

#203-208 is IVAC described as part of PVAP, please explain the percentages

# 227-230 pathogens must be written in italics and in graph 3 as well.

Author Response

Dear reviewer,

I sincerely thank you for your thorough work in pointing out some weaknesses in the manuscript and some points where the original text was unclear. The authors agreed to all suggestions.

We modified the text and graphs as prompted, and we are confident that the paper is remarkably improved.

The manuscript was amended additionally following the suggestions of reviewer 2

Best regards

Emanuele Russo

Comments and Suggestions for Authors

The author performed a retrospective analysis of infection-related ventilator-associated complications in critically ill patients with trauma

comments

#1 Abstract

why is the independent relationship between ICU outcomes and mortality highlighted in the abstract since this is not the main aim of this report

  • Correct observation. We deleted the sentence

#2 please state in the results (abstract) all the statistically significant results you found

  • Correct observation; we added significant results in the abstract

#3 The conclusions (abstract) are too vague. Please state explicitly

  • we rephrased the paragraph according to the reviewers' suggestions

#4 line 65-66 should combine with eprevious paragraph

  • Done

#5 line97-99 should combine with the previous paragraph

  • We deleted the period as it was redundant

#103-110 please state in this paragraph that you combined PVAP with IVAC

  • Thank you, extremely relevant suggestion; done

#131-132 please combine with previous paragraph

  • Done

# 163-170 Do the total number of patients described make the total of n=41?

  • We clarified that the finding refers to the population of the 88 selected cases

#203-208 is IVAC described as part of PVAP, please explain the percentages

  • Done

# 227-230 pathogens must be written in italics and in graph 3 as well.

  • Done

Reviewer 2 Report

In this retrospective cohort study the authors describe: 

  1. The prevalence of IVAC and PVAP among trauma patients who required mechanical ventilation for at least 4 days;
  2. The differences between those patients who developed IVAC and PVAP and those who did not;
  3. The pathogens that were responsible for the infection.

The cohort included 88 trauma patients who were ventilated for ≥96h. Twenty-six of them (29.5%) developed IVAC/PVAP. Most of these patients (21/26) had PVAP. The most common pathogens isolated from the respiratory tract were Staphylococcus aureus, E.coli, and Hemophilus influenza. 5/26 patients had resistant bacteria and only 7/26 had MDR bacteria. 

Patients who developed VAC/PVAP had higher ISS, slightly lower GCS, lower AIS of the abdomen, and more complicated clinical course, including ICU LOS, MV duration, and tracheostomy rate. The mortality rate is not mentioned (although it should be reported). 

The main conclusions of the authors were:

  1. Trauma patients have a low prevalence of resistant bacteria
  2. There is a low incidence of systemic involvement in trauma patients with IVAC/PVAP
  3. There are significant differences between MV duration and ICU LOS between patients who developed IVAC/PVAP and those who did not develop this complication. Both might represent risk factors for IVAC/PVAP or may be a consequence of IVAC/PVAP. 

The topic is interesting and important. The manuscript is well organized, however, it is slightly difficult to read and follow. Some language polishing is also required.

Major comments:

  1. As the authors state, the main limitation of this study is the small sample size. It prevented them from making any conclusion regarding the association between IVAC/PVAP and clinical outcomes as well as clarifying the risk factors for the development of IVAC/PVAP. These two cardinal aspects are not addressed by the authors. I would strongly recommend the authors increase the sample size and perform a multivariate analysis which will enable them to evaluate the risk factors for IVAC/PVAP and the association between these complications and the clinical course of the patients. 
  2. The low prevalence of resistant bacteria should be analyzed in the concept of the prevalence of resistant organisms in the hospital and the risk factors of the patients for antibiotic resistance (these are well described in previous studies). Both these aspects should be addressed by the authors. Without this information, the reader cannot know if the prevalence described in this cohort is high or low. 
  3. Trauma patients transferred to another hospital before ICU discharge should be excluded from the study as we can not know if they developed IVAC/PVAP or not.  

Specific comments:

ABSTRACT:

Please clarify the study design- as far as I understand it is a retrospective cohort study.

The Introduction should be shortened, and the Results section should include specific results rather than general statements. In addition the inclusion criteria (adult patients, severe trauma- ISS>15???, mechanical ventilation for at least 4 days)

INTRODUCTION:

This section is too long and difficult to follow. I would recommend shortening it and moving some content into the Methods (such as IVAC/PVAP definitions) and Discussion sections. I think that the discussion regarding TBI and the risk of VAP is irrelevant to this section. 

Ref #1: I failed to access the link provided. I believe that a more significant reference can be found. 

METHODS:

The inclusion criteria should be better defined- only severely injured patients (ISS>15) were included? 

As described in the results section, the relative risk of TBI patients developing IVAC/PVAP was calculated. Please define the statistic method used for this calculation in the Methods section. 

RESULTS:

Fig 1: please continue the flow chart until the outcome (IVAC/PVAP). 

Line 150: “43 patients developed infection or signs of infection during the ICU stay”- is not relevant here. 

Line 153: should be 41.5%

Table 1: p-value and percentages should be reported with “.” and not “,”. Kindly report the median or the mean according to the distribution of the values (normal or not). Reporting both median and mean is redundant. It is enough to report male or female gender. 

Lines 183-191: IQR should be reported as a range rather than a single number. 

Line 194: report p-value for RR.  

Graphs 1 and 4 do not add significant information to table 1 and the text and therefore can be omitted. 

Author Response

Dear reviewer,

sincerely thank you for your thoughtful work in analysing our manuscript and your valuable suggestions. The authors accepted all feasible suggestions.

We modified the text and graphs as prompted, some figures containing redundant information have been removed. We are confident that the paper is remarkably improved. A native English speaker revised the text. The manuscript is now fluent

Unfortunately, our dataset does not allow us to implement some research aspects; however, these shortcomings have been well highlighted in the Limitations section.

Below, in blue font formatted, are the remarks on each point.

Despite the limitations of the paper, we hope it will achieve acceptance for the special Issue "Healthcare Associated Infections" to which we were invited to contribute by the guest editor

Best regards

Comments and Suggestions for Authors

In this retrospective cohort study the authors describe: 

  1. The prevalence of IVAC and PVAP among trauma patients who required mechanical ventilation for at least 4 days;
  2. The differences between those patients who developed IVAC and PVAP and those who did not;
  3. The pathogens that were responsible for the infection.

The cohort included 88 trauma patients who were ventilated for ≥96h. Twenty-six of them (29.5%) developed IVAC/PVAP. Most of these patients (21/26) had PVAP. The most common pathogens isolated from the respiratory tract were Staphylococcus aureus, E.coli, and Hemophilus influenza. 5/26 patients had resistant bacteria and only 7/26 had MDR bacteria. 

Patients who developed VAC/PVAP had higher ISS, slightly lower GCS, lower AIS of the abdomen, and more complicated clinical course, including ICU LOS, MV duration, and tracheostomy rate. The mortality rate is not mentioned (although it should be reported). 

The main conclusions of the authors were:

  1. Trauma patients have a low prevalence of resistant bacteria
  2. There is a low incidence of systemic involvement in trauma patients with IVAC/PVAP
  3. There are significant differences between MV duration and ICU LOS between patients who developed IVAC/PVAP and those who did not develop this complication. Both might represent risk factors for IVAC/PVAP or may be a consequence of IVAC/PVAP. 

The topic is interesting and important. The manuscript is well organized, however, it is slightly difficult to read and follow. Some language polishing is also required.

Major comments:

  1. As the authors state, the main limitation of this study is the small sample size. It prevented them from making any conclusion regarding the association between IVAC/PVAP and clinical outcomes as well as clarifying the risk factors for the development of IVAC/PVAP. These two cardinal aspects are not addressed by the authors. I would strongly recommend the authors increase the sample size and perform a multivariate analysis which will enable them to evaluate the risk factors for IVAC/PVAP and the association between these complications and the clinical course of the patients. 

  • Unfortunately, with the data we hold, it is not possible to increase the sample size without losing accuracy. The absence of analogous robust data in the literature is precisely due to the difficulty of combining rigor and volume. We highlighted this issue at the first point in the "Limitations" section.

  1. The low prevalence of resistant bacteria should be analysed in the concept of the prevalence of resistant organisms in the hospital and the risk factors of the patients for antibiotic resistance (these are well described in previous studies). Both these aspects should be addressed by the authors. Without this information, the reader cannot know if the prevalence described in this cohort is high or low. 

  • Done; we addressed these points in the "Discussion"

  1. Trauma patients transferred to another hospital before ICU discharge should be excluded from the study as we cannot know if they developed IVAC/PVAP or not.  

  • We discussed the point in the "Limitations" section. The incidence of late infections in rehabilitation is beyond the scope of this paper as they exhibit different clinical characteristics and risk factors. The authors believe that removing these patients would further reduce the small sample size without gaining any particular advantage.

Specific comments:

ABSTRACT:

Please clarify the study design- as far as I understand it is a retrospective cohort study.

  • We clarified the point.

The Introduction should be shortened, and the Results section should include specific results rather than general statements. In addition, the inclusion criteria (adult patients, severe trauma- ISS>15???, mechanical ventilation for at least 4 days)

  • We agree with the remark, we shortened the introduction and reported the main results; We clarified the point of the inclusion criteria, ISS > 15 is not an inclusion criterion in our study; we added significant results in the abstract and rephrased the conclusion section according to the reviewers' suggestions

INTRODUCTION:

This section is too long and difficult to follow. I would recommend shortening it and moving some content into the Methods (such as IVAC/PVAP definitions) and Discussion sections. I think that the discussion regarding TBI and the risk of VAP is irrelevant to this section. 

  • We agree with the remark, we shortened the introduction and improved some references

Ref #1: I failed to access the link provided. I believe that a more significant reference can be found. 

  • Done; we have listed a stronger reference

METHODS:

The inclusion criteria should be better defined- only severely injured patients (ISS>15) were included? 

  • we clarified the point in the “Materials and Methods” section, ISS > 15 is not an inclusion criterium

As described in the results section, the relative risk of TBI patients developing IVAC/PVAP was calculated. Please define the statistic method used for this calculation in the Methods section. 

  • Done

RESULTS:

Fig 1: please continue the flow chart until the outcome (IVAC/PVAP). 

  • Done

Line 150: “43 patients developed infection or signs of infection during the ICU stay”- is not relevant here. 

  • We removed the sentence

Line 153: should be 41.5%

  • Done

Table 1: p-value and percentages should be reported with “.” and not “,”. Kindly report the median or the mean according to the distribution of the values (normal or not). Reporting both median and mean is redundant. It is enough to report male or female gender. 

  • We re-formatted the table according to the suggestions.

Lines 183-191: IQR should be reported as a range rather than a single number. 

  • We reported the difference between the 75th and 25th percentilesof the data; We believe both systems are correct; we preferred to avoid an excess of numbers to make text and tables readable (Upton, Graham; Cook, Ian (1996). Understanding Statistics. Oxford University Press. p. 55. ISBN 0-19-914391-9.)

Line 194: report p-value for RR.  

  • We reported the confidence interval as a measure of reliability of the result

Graphs 1 and 4 do not add significant information to table 1 and the text and therefore can be omitted. 

  • Done

Round 2

Reviewer 2 Report

The authors addressed all the suggested comments. Good luck.